# Observational cohort study with internal and external validation of a predictive tool for identification of children in need of hospital admission from the emergency department: the Paediatric Admission Guidance in the Emergency Department (PAGE) score

Andrew Rowland,[1,2] Sarah Cotterill,[3] Calvin Heal,[3] Natalie Garratt,[4] Tony Long,[1] Laura Jayne Bonnett ![ORCID],[5] Stephen Brown,[4] Steve Woby,[1,4] Damian Roland ![ORCID] [6,7]

► Prepublication history and additional materials for this paper is available online. To view these files, please visit the journal online (http://dx.doi.org/10.1136/bmjopen-2020-043864).

**Correspondence to**
Dr Damian Roland;
dr98@le.ac.uk

## ABSTRACT

**Objectives** To devise an assessment tool to aid discharge and admission decision-making in relation to children and young people in hospital urgent and emergency care facilities, and thereby improve the quality of care that patients receive, using a clinical prediction modelling approach.

**Design** Observational cohort study with internal and external validation of a predictive tool.

**Setting** Two general emergency departments (EDs) and an urgent care centre in the North of England.

**Participants** The eligibility criteria were children and young people 0–16 years of age who attended one of the three hospital sites within one National Health Service (NHS) organisation. Children were excluded if they opted out of the study, were brought to the ED following their death in the community or arrived in cardiac arrest when the heart rate and respiratory rate would be unmeasurable.

**Main outcome measures** Admission or discharge. A participant was defined as being admitted to hospital if they left the ED to enter the hospital for further assessment, (including being admitted to an observation and assessment unit or hospital ward), either on first presentation or with the same complaint within 7 days. Those who were not admitted were defined as having been discharged.

**Results** The study collected data on 36 365 participants. 15 328 participants were included in the final analysis cohort (21 045 observations) and 17 710 participants were included in the validation cohort (23 262 observations). There were 14 variables entered into the regression analysis. Of the 13 that remained in the final model, 10 were present in all 500 bootstraps. The resulting Paediatric Admission Guidance in the Emergency Department (PAGE) score demonstrated good internal validity. The C-index (area under the ROC) was 0.779 (95% CI 0.772 to 0.786).

**Conclusions** For units without the immediate availability of paediatricians the PAGE score can assist staff to

### Strengths and limitations of this study

► This study was novel in that it was based on data from children attending three non-tertiary (non-specialist) hospitals (n=36 365). Non specialist hospitals are where the majority of children are seen by emergency and urgent care practitioners in the National Health Service in England.

► The Paediatric Admission Guidance in the Emergency department score, was built from the clinical prediction model derived by both an internal and external data validation set.

► Admission rates varied between hospitals and while internal and external validation occurred the impact of prospective use to guide decision will need confirmation.

determine risk of admission. Cut-off values will need to be adjusted to local circumstance.

**Study protocol** The study protocol has been published in an open access journal: Riaz *et al* Refining and testing the diagnostic accuracy of an assessment tool (Pennine Acute Hospitals NHS Trust-Paediatric Observation Priority Score) to predict admission and discharge of children and young people who attend an ED: protocol for an observational study. BMC Pediatr 18, 303 (2018). https://doi.org/10.1186/s12887-018-1268-7.

**Trial registration number** The protocol has been published and the study registered (NIHR RfPB Grant: PB-PG-0815–20034; ClinicalTrials.gov:213469).

## INTRODUCTION

Attendances at emergency departments (EDs) by children and young people have risen year on year in the UK and are now in excess of 4 million per year.[1] The majority of

these attendances are at services where the professional contact is by a practitioner without formal paediatric training. Given that a large proportion of these children and young people are discharged safely, the delivery of emergency and acute care does not rely on direct paediatric review.

However, systems are needed to recognise children who require further observation and investigation. Triage processes are well established to determine the order in which patients should receive clinical review. Increasingly, a fuller assessment, including the patient's vital signs, is used to determine an acuity score to define risk of illness or deterioration. These scores, often termed Early Warning Scores (EWS), originated in adult in-patient environments where it was recognised that death or intensive care admission was often preceded by derangement of the patient's physiology.[2] These findings led to the development of paediatric EWSs (PEWS) which have been shown to have utility in the detection of deterioration, but not always in the prevention of mortality.[3] The face validity of such scores led to widespread utilisation and attempts to employ them in different emergency and prehospital environments from those where the scores were devised. Initial reviews demonstrated relatively poor performance,[4] in turn leading to the development of bespoke scores validated in EDs themselves.

There is no existing gold-standard outcome measure for the decision to admit or discharge a child or young person from the ED,[5] and the decision to admit is a complex one, which can vary between clinicians and hospitals. One EWS system, the Paediatric Observation Priority Score (POPS), recommended by the Intercollegiate Committee for Standards for Children and Young People in Emergency Care Settings[6] has shown initial promise in aiding recognition of unwell children but also aiding safe discharge decisions.[7 8]

A modified version of POPS, developed at The Pennine Acute Hospitals National Health Service (NHS) Trust (PAT-POPS), has been shown to have greater discriminatory power than an inpatient PEWS.[9] The majority of EWS systems are derived and validated at large teaching institutions ('tertiary centres') and so may not be applicable outside tertiary centres.[10] In this study, PAT-POPS was used as the initial starting point to derive and validate a score to aid disposition decisions in non-tertiary centres. This would have external validity in the majority of locations where children and young people are assessed in the UK and potentially beyond.

## METHODS

A published protocol is available[11] and only the core components of the methods are summarised in this paper. In this paper, 'participant' means a patient recruited into the study at one of the study sites.

## Patient and public involvement

The study was supported by a patient advisory group which provided input to the programme of research. This patient advisory group met with author TL during the study. Patients partnered with the study team for the design of the study, the informational material to support the opt-out consent process. At the end of the study twenty-one parents were consulted in one of the EDs on how the cut point might be made. Most found this a particularly difficult task, hovering around the centre of the scale. When the notion was introduced of a supporting community service that would know about their child and for which they had a contact number if worried, a remarkable difference was observed. All but two (both fathers) considered correct discharge to be more important than correct admission, and, overall, most thought that about 60%–70% correct discharge was optimal. Two mothers would be sufficiently confident for a much higher specificity, being prepared to discharge 90%–95% patients correctly.

## Sample population

The study population was recruited consecutively. Data collection was prospective over a whole year (1 March 2018–28 February 2019) to avoid the effects of bias from seasonal variability. The eligibility criteria were children and young people 0–16 years of age who attended one of three hospital sites within one NHS Trust (NHS organisation) in Greater Manchester, UK. Children were excluded if they opted out of the study, were brought to the ED following their death in the community, or arrived in cardiac arrest when the heart rate and respiratory rate would be unmeasurable.

## Admission

A participant was defined as being admitted to hospital if they left the ED to enter the hospital for further assessment, (including observation and assessment unit or hospital ward), either on first presentation or with the same complaint within 7 days. This was approved by the study patient and public involvement group, which saw admission and discharge in such terms, and by discussion with three ED doctors.

The decision to admit the participant was made by a clinician (either a doctor or a nurse practitioner). They followed existing guidelines, using usual methods of clinical judgement, and were blinded to the score as this had not yet been derived at the point of participant assessment (but had access to all vital signs and observations normally collected). Admission data from all three hospitals was accessed from existing NHS trust electronic systems.

## Predictors

All of the variables in the modified PAT-POPS plus additional variables included in other scores were considered for inclusion in the Paediatric Admission Guidance in the Emergency Department (PAGE) score (table 1).

**Table 1** Characteristics of the sample population at each site (who met the inclusion criteria)

| Variable | Site 1 (emergency department) (Development dataset) Visits=21480 Participants=15484 | | | Site 2 (emergency department) Visits=9972 Participants=7817 | | | Site 3 (urgent care centre) Visits=13429 Participants=9971 | | |
|---|---|---|---|---|---|---|---|---|---|
| | Summary | Min/max | Missing (%) | Summary | Min/max | Missing (%) | Summary | Min/max | Missing (%) |
| Age in years (median, IQR) | 3 (1–8) | 0/15 | 0 | 7 (3–12) | 0/15 | 0 | 6 (2–10) | 0/15 | 0 |
| Gender | | | 0 | | | 0 | | | 0 |
| Female | 9384 (44%) | | | 4486 (45%) | | | 6131 (54%) | | |
| Ethnicity | | | 0 | | | 0 | | | 0 |
| White British | 11162 (52%) | | | 7081 (71%) | | | 7046 (52%) | | |
| Irish+Gypsy or Irish Traveller | 34 (0%) | | | 26 (0%) | | | 61 (0%) | | |
| Other white | 569 (3%) | | | 298 (3%) | | | 369 (3%) | | |
| White and black Caribbean | 158 (1%) | | | 89 (1%) | | | 51 (0%) | | |
| White and black African | 121 (1%) | | | 53 (1%) | | | 80 (1%) | | |
| White and Asian | 255 (1%) | | | 118 (1%) | | | 217 (2%) | | |
| Other mixed | 294 (1%) | | | 107 (1%) | | | 244 (2%) | | |
| Indian | 75 (0%) | | | 45 (0%) | | | 51 (0%) | | |
| Pakistani | 4245 (20%) | | | 1165 (12%) | | | 61 (0%) | | |
| Bangladeshi | 2060 (10%) | | | 36 (0% | | | 524 (4%) | | |
| Chinese | 46 (0%) | | | 42 (0%) | | | 29 (0%) | | |
| Other Asian+Asian/Asian British | 302 (1%) | | | 202 (2%) | | | 213 (2%) | | |
| African | 374 (2%) | | | 114 (1%) | | | 242 (2%) | | |
| Caribbean | 24 (0%) | | | 8 (0%) | | | 21 (0%) | | |
| Other Black+Black/African/Caribbean/Black British | 149 (1%) | | | 54 (1%) | | | 168 (1%) | | |
| Unknown/not stated | 1010 (5%) | | | 270 (3%) | | | 169 (1%) | | |
| Heart rate (mean/SD) | 121.6 (30.5) | | 11 | 104.5 (26.1) | | 4 | 109.7 (27.0) | | 2 |
| Temperature (median/IQR) | 36.9 (36.5 to 37.4) | 32.6/41.7 | 10 | 36.7 (36.4 to 37.1) | 34.4/41.1 | 3 | 36.7 (36.4 to 37.1) | 35.0/41.2 | 1 |
| Respiratory rate | 26 (22 to 32) | 10/90* | 10 | 22 (20 to 26) | Dec-88 | 3 | 24 (20 to 26) | Dec-80 | 2 |
| Oxygen saturation | 99 (98 to 100) | 63/100 | 10 | 99 (98 to 100) | 70/100 | 4 | 99 (99 to 100) | 67/100 | 2 |
| Requires supplement oxygen | 386 (2%) | | 0 | 31 (0%) | | 0 | 99 (1%) | | 0 |
| Breathing | | | 2 | | | 0 | | | 0 |
| None of the above | 19678 (93%) | | | 9713 (98%) | | | 13038 (98%) | | |

Continued

**Table 1** Continued

| Variable | Site 1 (emergency department) (Development dataset) Visits=21 480 Participants=15 484 | | | Site 2 (emergency department) Visits=9972 Participants=7817 | | | Site 3 (urgent care centre) Visits=13 429 Participants=9971 | | |
|---|---|---|---|---|---|---|---|---|---|
| | Summary | Min/max | Missing (%) | Summary | Min/max | Missing (%) | Summary | Min/max | Missing (%) |
| Audible grunt | 52 (0%) | | | 9 (0%) | | | 25 (0%) | | |
| Wheeze | 709 (3%) | | | 135 (1%) | | | 190 (1%) | | |
| Stridor | 52 (0%) | | | 6 (0%) | | | 9 (0%) | | |
| Tracheal tug | 624 (3%) | | | 68 (1%) | | | 102 (1%) | | |
| Recession | | | 0 | | | 0 | | | 0 |
| No recession | 18 757 (87%) | | | 9748 (98%) | | | 12 900 (96%) | | |
| Mild recession | 1764 (8%) | | | 153 (2%) | | | 338 (3%) | | |
| Moderate | 743 (3%) | | | 43 (0%) | | | 114 (1%) | | |
| Severe | 213 (1%) | | | 27 (0%) | | | 75 (1%) | | |
| Responsiveness | | | 0 | | | 0 | | | 0 |
| Alert | 21 044 (98%) | | | 9775 (98%) | | | 13 252 (99%) | | |
| Responds to pain | 53 (0%) | | | 10 (0%) | | | 5 (0%) | | |
| Responds to voice | 126 (1%) | | | 15 (0%) | | | 19 (0%) | | |
| Unresponsive | 253 (1%) | | | 171 (2%) | | | 151 (1%) | | |
| Nurse judgement | | | 0 | | | 0 | | | 0 |
| No concern | 6476 (30%) | | | 6894 (69%) | | | 9888 (74%) | | |
| Low level concern | 11 645 (54%) | | | 2756 (28%) | | | 3268 (24%) | | |
| Child looks unwell/high concern | 3355 (16%) | | | 321 (3%) | | | 271 (2%) | | |
| Behaviour | | | 0 | | | 0 | | | 0 |
| Normal for age | 20 349 (95%) | | | 9720 (97%) | | | 13 105 (98%) | | |
| Agitated | 356 (2%) | | | 105 (1%) | | | 121 (1%) | | |
| Floppy | 138 (1%) | | | 22 (0%) | | | 23 (0%) | | |
| Inappropriate | 54 (0%) | | | 9 (0%) | | | 8 (0%) | | |
| Listless | 579 (3%) | | | 115 (1%) | | | 170 (1%) | | |
| Multi-morbidity | 749 (3%) | | 0 | 148 (1%) | | 0 | 551 (4%) | | 0 |
| Arrived by ambulance | 4539 (21%) | | 0 | 20 (0%) | | 0 | 50 (0%) | | 0 |
| Admitted now or next 7 days | 6810 (32%) | | 0 | 600 (6%) | | 0 | 1009 (8%) | | 0 |

*This was the highest value that could be entered.

Other data collection included reason for attendance at the ED, diagnosis, death in the ED, children leaving the ED before admission decision, children's characteristics (age, gender and ethnicity), investigated deaths and serious incidents. These variables were chosen as in the authors clinical experience they may have dependency on the decision to admit but have also not be formally studied in previous analyses.[4]

## Sample size

In the protocol, it was estimated that 9000 children were needed for the development of the prediction model and 7000 children in the independent validation. The minimum sample size needed to do both analyses was therefore estimated as 16 000 children. More data than needed was allowed for (and this approach was granted at ethics review) due to the need to collect data for a full year to capture seasonal variation in childhood illness and injury. Intermittent data collection would not help implementation of the tool and would have required the employment of specific staff for the project, which would have been significantly more costly.

## Analysis

Analysis was conducted in StataMP V.14[12] using two-sided 95% CIs and the 5% significance level. Analysis was reported according to the Transparent Reporting of a multivariable prediction model for Individual Prognosis Or Diagnosis (TRIPOD[13]) and Standards for Reporting of Diagnostic Accuracy[14] reporting guidelines. A flow diagram summarises participant opt-out, recruitment and data collection, by site (figure 1).

Final analysis was undertaken after all data had been entered into the database, and the database had been cleaned and locked. Children were excluded from the analysis if the outcome variable (admission) was missing or if all the independent variables were missing. For variables with over 4% rates of missingness data was imputed using hot-deck imputation.[15] This involves stratifying

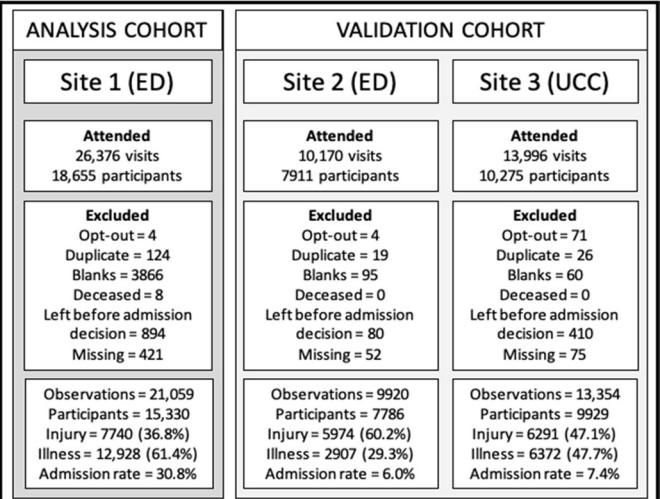

**Figure 1** Development of the dataset. ED, emergency department. UCC, Urgent Care Centre

participants by key predictive variables (injury/illness status, admission status and age) and replacing missing values with those of a participant from the same strata. Participants were described with respect to the variables in the model, both overall and by site, reported as number (%) for categorical variables; mean (SD, min, max) for normally distributed variables; median (IQR, min, max) for other numeric variables.

## Stage 1: model development

Children from one hospital site were utilised for the model development. Logistic regression models were developed with hospital admission as the outcome and including all candidate variables. Due to several of the variables being non-linearly associated with the outcome a closed test procedure was used to determine the best functional form of each continuous variable and concurrently whether it should be included. This involved starting with a model that included all potential predictors and testing the best fitting fractional polynomial form of each continuous variable individually.[16] Variables were removed from the model according to p values with any less than 0.10 being excluded. For categorical variables, if at least one category was significant all were included initially. Quality of the data and risk of bias was assessed using the Prediction model Risk Of Bias ASsessment Tool (PROBAST).[17]

## Stage 2: internal validation

The calibration score and calibration slope were examined to assess how well the predictions from the model matched the data and calibration plots were used to compare agreement between predicted and observed injury and illness. Discrimination was also considered, to measure how well the model separated between individuals who were admitted and those who were not (C-statistic, which is equivalent to area under the receiver operating characteristic (ROC) curve.

The Brier score has been reported. Internal validation was performed by applying the original model to 500 bootstrapped samples. The discrimination and calibration performance of the model in each of the bootstrap samples was compared with the model fitted to the original data to provide a measure of optimism. Due to the large sample size, it was not necessary to adjust for the optimism.[18] The inclusion/exclusion of any predictors which featured in the selected model but only rarely across the bootstrap samples (or vice versa) was noted.

The output of stage 1 and 2 was the PAGE score which aimed to predict hospital admission, based on the weight of each predictor.

## Stage 3: external validation
### Assessing model performance

External validation was undertaken using data from two other hospitals. The developed model was applied to each external dataset, and calibration and discrimination measures were reported, as above.

## Recalibration

After looking at both the development and the external datasets, and with clinical opinion, consideration was given to which dataset most closely represents the majority of UK EDs at which children attend, and accordingly whether or not to recalibrate the model based on one of the external validation datasets.

## Stage 4: developing a point scoring system

The model's regression coefficients were used to assign integer points to each level of each risk factor, and a reference table of risk per possible points total was produced, following established guidelines.[19] Together these provide a clinically useful score. By applying the points scores to the development dataset it was possible to calculate the sensitivity, specificity, positive and negative likelihood ratios of PAGE (index test) in predicting admission (reference test) with 95% CIs. A cross tabulation of the results of the index test by the results of the reference test was reported, including indeterminate and missing results.

## Consensus meeting

A meeting was held to examine the statistical data, and to agree which cut points of the PAGE score were most suitable to predict (1) safe admission decision and (2) safe discharge decision, including consideration of what weight to give to sensitivity and specificity in making the decision. All of our research team, plus paediatric ED clinicians and an independent methodologist, were invited to attend.

## Stage 5: further analyses

The usefulness of the PAGE score was assessed by calculating the sensitivity, specificity, positive and negative likelihood ratios at the chosen cut-points, to predict admission and discharge.

## Research governance

An opt-out consent design was approved, and there were no plans to disseminate the results to participants.

## RESULTS

The unit of analysis was a visit by a child to the ED. Figure 1 shows the numbers and reasons for exclusions of visits (and participants) from the dataset. Children were omitted because they opted out of the study (n=4, site 1), died in the ED (n=8, Site 1) or left the ED before an admission decision was made (n=892, site 1).

Duplicate entries where participants had more than one entry with the same date and time (n=199, site 1) were deleted.

## Missing visits

There were two types of visit that were excluded completely from the development dataset due to extensive missing data. The first (marked as 'blanks' in figure 1) were missing all data except for patient demographics. After

data collection was completed, we discovered that site 1 had a two-tier system which funnels off particularly 'well' children into a separate nurse-led service where no further study information was recorded (n=3865, site 1). The second were visits where data were missing for enough clinical measures and judgements to prevent imputation and inclusion in the analysis This was likely due to them being measurements that would not routinely be taken for certain types of attendances, such as minor injuries (n=421, site 1).

## Missing variables

Most variables had very low rates of missingness (0%–2%) and no imputation was required. Four variables: heart rate, temperature, respiratory rates and oxygen saturation had higher rates, which varied by site: site 1 (8%–11%), site 2 (2%–4%) and site 3 (1%–2%). Reasons for the missingness, and differences in missingness by site, are not clear. Values were imputed using hot-deck imputation.[15]

## Descriptive statistics/summary of population

Table 1 summarises the demographic and medical characteristics of the sample population at each site. This comprises all participants who met the inclusion criteria. The admission rate of site 1 (the development dataset) was 32% compared with 6% at site 2% and 8% at site 3. Compared with the external validation sites, children in the development dataset were younger, showed signs of being more unwell (higher heart rate, more prevalence of recession and breathing abnormalities) and were more likely to show cause for concern for the nurses. Temperature, oxygen saturation, responsiveness and behaviour were very similar across the sites.

The form of variables and their suitability for inclusion in the model were considered and adjusted as necessary. Categorical variables were recoded to avoid any small categories. Abnormal breathing was recategorised as 'normal' and 'abnormal'. 'severe' and 'moderate' recession were amalgamated. Two continuous variables (heart rate and oxygen saturation) were categorised as in their continuous form we were unable to model them adequately while keeping the final score readily usable in clinical practice.

## Regression results

The final multivariable logistic regression model is shown in table 2. Responsiveness was omitted from the final model, because once the other variables were accounted for, it had no effect on admission. All other variables were retained in the final model and had a statistically significant relationship with admission except for the 'under 75 bpm' heart rate category.

The younger the child the more likely admission is and the effect is much stronger for much younger participants. Compared with those aged 4–16 years old, those aged 0–1 months had an increase in admission odds of 1.880 (95% CI 1.876 to 1.885). Low oxygen saturation was the strongest predictor of admission in the study and

**Table 2** Final regression model

| Variable | OR | 95% CI low | 95% CI high | P value |
|---|---|---|---|---|
| Age | | | | <0.001 |
| 0–1 months | 1.88 | 1.876 | 1.885 | |
| 2–5 months | 1.279 | 1.276 | 1.282 | |
| 6–11 months | 1.125 | 1.122 | 1.282 | |
| 1–3 years | 1.031 | 1.029 | 1.034 | |
| 4–16 years | Reference | | | |
| Heart rate | | | | |
| <75 | 0.959 | 0.807 | 1.141 | 0.638 |
| 75–125 | Reference | | | |
| >125 | 1.422 | 1.303 | 1.552 | <0.001 |
| Temperature (degrees Celsius) | | | | <0.001 |
| <38 | Reference | | | |
| 38–39 | 1.365 | 1.307 | 1.424 | |
| >39 | 1.763 | 1.689 | 1.84 | |
| Respiratory rate | 1.029 | 1.023 | 1.035 | <0.001 |
| Oxygen saturation | | | | <0.001 |
| 95–100 | Reference | | | |
| 90–94 | 3.381 | 2.654 | 4.305 | |
| <90 | 5.477 | 2.953 | 10.157 | |
| Requires supplementary oxygen | 1.872 | 1.437 | 2.44 | <0.001 |
| Breathing | | | | |
| Normal | Reference | | | |
| Abnormal | 1.681 | 1.423 | 1.985 | <0.001 |
| Recession | | | | |
| None | Reference | | | |
| Mild | 1.281 | 1.117 | 1.469 | <0.001 |
| Moderate/severe | 1.484 | 1.183 | 1.863 | 0.001 |
| Behaviour category | | | | |
| Normal | Reference | | | |
| Agitated | 1.353 | 1.043 | 1.754 | 0.023 |
| Floppy | 2.046 | 1.371 | 3.054 | <0.001 |
| Listless | 1.498 | 1.217 | 1.843 | <0.001 |
| Nurse judgement | | | | |
| No concern | Reference | | | |
| Low level concern | 1.361 | 1.258 | 1.473 | <0.001 |
| High concern | 2.677 | 2.388 | 3.002 | <0.001 |
| Multimorbidity | 1.937 | 1.599 | 2.346 | <0.001 |
| Arrived by ambulance | 2.416 | 2.223 | 2.625 | <0.001 |
| Advised by medical professional | 2.159 | 1.975 | 2.361 | <0.001 |
| Constant | 0.144 | 0.133 | 0.156 | <0.001 |

nurse judgement was strongly associated with admission. Variables that were introduced in this study as additional to the existing PAT-POPS score were multimorbidity, (1.94 (95% CI 1.60 to 2.35)), arrival by ambulance (2.42 (95% CI 2.22 to 2.63)) and being advised to attend by a medical professional (2.16 (95% CI 1.97 to 2.36)).

### Internal validation

As would be expected from such a large sample size, the model demonstrated good internal validity. The discrimination (C-index) of the model in the original dataset was 0.779 (95% CI 0.772 to 0.786) and the mean in the bootstrap samples was 0.779 (95% CI 0.767 to 0.789)

**Table 3** External validation results

| Site | Briers score | Calibration slope | CITL | E/O | C-index |
|------|-------------|-------------------|------|-----|---------|
| 2 | 0.065 | 0.986 (0.897 to 1.076) | −1.219 (−1.306 to −1.132) | 2.673 | 0.763 (0.742 to 0.783) |
| 3 | 0.073 | 1.075 (1.002 to 1.149) | −1.055 (−1.123 to −0.987) | 2.275 | 0.753 (0.737 to 0.770) |

CITL, Calibration in the Large E/O, Expected/Observed

suggesting no model optimism. Full calibration and discrimination results are reported in the online supplemental file.

### Calibration

The Brier score was 0.1664, the calibration slope was 1.000 and calibration-in-the-large was −0.000. The calibration plot indicates that predictions from the model match the data very well for low to moderate risk individuals. Above this, the model slightly overpredicts the probability of admission.

### External validation

The external validation was undertaken in each of the two external datasets separately (table 3).

These results indicate that the model transportability is good overall, with only slight over/underfitting present at each of the external validation sites and C-indexes (0.763 and 753 for sites 2 and 3, respectively), only modestly smaller than were found in the original model. It is clear however that the model may over-estimate the probability of admission in external populations, probably owing to the unusually high admission rate of the development site (site 1).

Of the three sites, site 1 is most typical of UK EDs in respect of annual attendances and size of unit. This being the case, recalibration of the model so it performed better in the two external datasets was not necessary. The original model was carried forward to the stage of developing the point scoring system.

### Derivation of PAGE score

The PAGE score has a minimum value of 0 and a maximum of 30 (table 4). The characteristic that yields the largest numbers of points is having a lower than normal oxygen saturation, worth four or five points depending on severity. Age, against clinical expectation, seemed only predictive of admission when the participant was younger than 6 months.

The sensitivity and specificity of each potential cut-point of the tool is shown in table 5. As the development of the point-scoring system was effectively a simplification of the regression model, we expected a relative decrease in its predictive power. The difference in the discrimination of the regression model and the resulting point scoring tool however proved to be minimal (figure 2): a drop from 0.779 (95% CI 0.772 to 785) to 0.775 (95% CI 0.768 to 0.781).

The cut-point for PAGE that most closely aligns with the sensitivity and specificity of the chosen original PAT-POPS

cut-off is a score of 7 or more. This yields a sensitivity of 48.21% and specificity of 87.61%. Based on the preferences of those at the consensus meeting the chosen cut-off for their organisations would be either six or seven

**Table 4** PAGE point scoring tool

| Variable | Category | Point(s) |
|----------|----------|----------|
| Age (months) | 0–1 | 2 |
| | 2–5 | 1 |
| | ≥6 | 0 |
| Heart rate (BPM) | ≤125 (ie, 0–125) | 0 |
| | >125 (ie, 126 and above) | 1 |
| Respiratory rate | 0–25 | 0 |
| | 26–60 | 2 |
| | >60 | 3 |
| Temperature (degrees Celsius) | <38 | 0 |
| | 38–39 | 1 |
| | >39 | 2 |
| Oxygen saturation (%) | >94 | 0 |
| | 90–94 | 4 |
| | <90 | 5 |
| Requires supplementary oxygen | No | 0 |
| | Yes | 2 |
| Breathing | Normal | 0 |
| | Abnormal | 2 |
| Recession | No recession | 0 |
| | Any recession | 1 |
| Behaviour | Normal | 0 |
| | Agitated or Listless | 1 |
| | Floppy | 2 |
| Nurse judgement | No concern | 0 |
| | Low level concern | 1 |
| | High concern | 3 |
| Multi-morbidity | No | 0 |
| | Yes | 2 |
| Arrived by ambulance | No | 0 |
| | Yes | 3 |
| Advised by medical professional to attend | No | 0 |
| | Yes | 2 |

PAGE, Paediatric Admission Guidance in the Emergency Department.

| Table 5 | Sensitivity and specificity for each cut-point of PAGE | | | | | |
|---|---|---|---|---|---|---|
| Cut-point for admission decision | Sensitivity (95% CI) | Specificity (95% CI) | Likelihood ratio+ | Likelihood ratio− | Positive predictive value | Negative predictive value |
| ≥0 | 100% | 0% | 1.0000 | – | – | – |
| ≥1 | 96.83 (96.38 to 97.23) | 15.80 (15.22 to 16.41) | 1.1501 | 0.2010 | 34.8 (34.2 to 35.4) | 91.5 (91.1 to 91.9) |
| ≥2 | 91.13 (90.42 to 91.79) | 36.34 (35.56 to 37.13) | 1.4315 | 0.2441 | 39.9 (39.3 to 40.6) | 89.8 (89.4 to 90.2) |
| ≥3 | 87.70 (86.89 to 88.46) | 45.84 (45.03 to 46.65) | 1.6192 | 0.2684 | 42.9 (42.3 to 43.6) | 88.9 (88.5 to 89.3) |
| ≥4 | 78.87 (77.90 to 79.81) | 61.03 (62.23 to 63.82) | 2.1330 | 0.3353 | 48.5 (47.8 to 49.1) | 85.7 (85.2 to 86.2) |
| ≥5 | 68.61 (67.49 to 69.70) | 73.69 (72.97 to 74.40) | 2.6074 | 0.4261 | 54.8 (54.1 to 55.4) | 83.5 (83.0 to 84.0) |
| ≥6 | 59.47 (58.29 to 60.64) | 80.86 (80.21 to 81.49) | 3.1070 | 0.5012 | 59.1 (58.4 to 59.7) | 81.1 (80.6 to 81.7) |
| ≥7 | 48.56 (47.37 to 49.76) | 87.89 (87.03 to 88.11) | 3.9099 | 0.5873 | 64.5 (63.8 to 65.1) | 78.6 (78.0 to 79.1) |
| ≥8 | 38.88 (37.73 to 40.06) | 92.20 (91.75 to 92.62) | 4.9819 | 0.6629 | 69.8 (69.2 to 70.4) | 76.5 (75.9 to 77.0) |
| ≥9 | 29.84 (28.76 to 30.94) | 95.34 (94.99 to 95.68) | 6.4089 | 0.7359 | 74.8 (74.3 to 75.4) | 74.5 (74.0 to 75.1) |
| ≥10 | 22.11 (21.14 to 23.12) | 97.43 (97.16 to 97.68) | 8.6053 | 0.7994 | 80.0 (79.4 to 80.5) | 72.9 (72.3 to 73.5) |
| ≥11 | 16.36 (15.49 to 17.26) | 98.56 (98.35 to 98.75) | 11.3733 | 0.8486 | 84.1 (83.6 to 84.6) | 71.7 (71.1 to 72.3) |
| ≥12 | 12.03 (11.27 to 12.83) | 99.28 (99.12 to 99.41) | 16.6441 | 0.8861 | 88.5 (88.1 to 89.0) | 70.9 (70.3 to 71.5) |
| ≥13 | 8.33 (7.69 to 9.01) | 99.60 (99.48 to 99.69) | 20.7021 | 0.9204 | 90.6 (90.2 to 91.0) | 70.1 (69.5 to 70.7) |
| ≥14 | 5.49 (4.97 to 6.07) | 99.79 (99.70 to 99.85) | 25.9892 | 0.9471 | 92.4 (92.0 to 92.7) | 69.5 (68.9 to 70.1) |
| ≥15 | 3.25 (2.84 to 3.70) | 99.85 (99.77 to 99.90) | 21.6398 | 0.9690 | 91.0 (90.6 to 91.3) | 69.0 (68.4 to 69.6) |
| ≥16 | 2.14 (1.82 to 2.52) | 99.90 (99.83 to 99.94) | 20.9674 | 0.9796 | 90.7 (90.3 to 91.1) | 68.7 (68.1 to 69.4) |
| ≥17 | 1.12 (0.89 to 1.40) | 99.97 (99.92 to 99.99) | 32.7436 | 0.9867 | 93.8 (93.5 to 94.2) | 68.5 (67.9 to 69.2) |
| ≥18 | 0.65 (0.48 to 0.88) | 100 (99.97 to 100) | – | 0.9935 | 100.0 (100.0 to 100.0) | 68.4 (67.8 to 69.1) |
| ≥19 | 0.34 (0.22 to 0.52) | 100 (99.97 to 100) | – | 0.9966 | 100.0 (100.0 to 100.0) | 68.4 (67.8 to 69.0) |
| ≥20 | 0.12 (0.06 to 0.24) | 100 (99.97 to 100) | – | 0.9988 | 100.0 (100.0 to 100.0) | 68.3 (67.7 to 68.9) |
| ≥21 | 0.04 (0.01 to 0.14) | 100 (99.97 to 100) | – | 0.9996 | 100.0 (100.0 to 100.0) | 68.3 (67.7 to 68.9) |
| ≥22 | 0.02 (0.00 to 0.10) | 100 (99.97 to 100) | – | 0.9996 | 100.0 (100.0 to 100.0) | 68.3 (67.7 to 68.9) |
| ≥23 | 0 | 100 | – | 1.0000 | 100.0 (100.0 to 100.0) | 68.3 (67.7 to 68.9) |

PAGE, Paediatric Admission Guidance in the Emergency Department.

points on the basis that consensus meeting attendees indicated a preference for giving more weight to higher specificity (ensuring the right children were discharged)

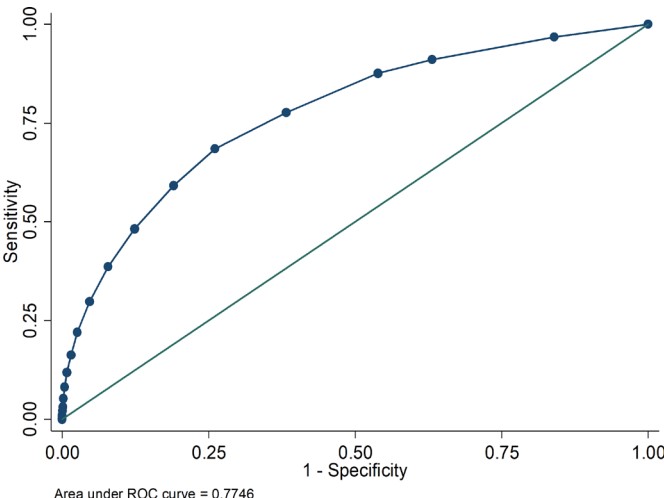

**Figure 2** Area under receiver operating characteristic (ROC) curve V.2.

than sensitivity. It is, of course, a matter for healthcare organisations to determine their own cut-off based on local service arrangements, including what community services may be available to those patients who are discharged.

## DISCUSSION

In this study of over 30 000 children the PAGE score was derived. This is the first study of its kind in non-tertiary EDs and therefore should have good external validity in the majority of locations in which children are treated and in which paediatricians are not immediately available. While the development site (site 1), a type 1 ED with trauma unit status, had a high admission rate, it is noted that the national average is 13.8% (CIs unavailable).[20] At site 1, the participants presenting with minor injury are streamed into an emergency nurse practitioner stream which artificially increases the apparent admission rate. When tested on the sites with admission rates of 6% and 8% the score still performed well, highlighting that the score may have utility in organisations with a range of

admission rates. The AUROC curve 0.779 (95% CI 0.772 to 0.785) performing better than any other previously published similar tool.[21]

The PAGE score should not be considered a typical EWS as it has been derived for the sole purpose of determining risk of admission, rather than risk of deterioration. A number of the components are temporally fixed (ie, arrival by ambulance and referred by healthcare professional) and will not change, so the PAGE score is not a sequential scoring tool, either. However, as all the components are commonly utilised it provides an easy to measure score at initial assessment that can guide the emergency pathway utilisation for the child or young person. It is important to note that age is a component of the score itself (ie, there are not separate scores for each age group). For those used to traditional EWS approaches this may cause uncertainty as the heart rate cut-off appears low for many young infants (>125 gives a score of 1 across all ages). However, it is important to remember that this is a population-derived tool and therefore at a patient level some features may not seem clinically relevant. This is why, importantly, it will be necessary for local departments to derive their own cut points for the overall PAGE score as where they are set depends on what the local services can offer both in the hospital and in the community. We also suggest in future validation it will be possible to asses it's feasibility as part of an electronic health record which will reduce the burden on healthcare staff calculating the total score.

The derivation of score revealed some interesting results in relation to the relative importance of the different components. While nurse judgement was the categorical variable with the strongest association with admission this was only marginal. The CI for 'floppy' means that this may also be a strong predictor, but there were insufficient numbers of participants presenting with this characteristic to be able to demonstrate this. Although from a continuous variable, having an oxygen saturation below 90 was a much stronger predictor of admittance, which is in keeping with known clinical practice. However, in a group of objective measures, it is perhaps surprising one of the most subjective, judgement, was one of the strongest associations. This may reflect the important predictive information that cannot or will not be obtained through clinical measurements and is clearly an area worthy of further study.

The PAGE score did not assess individual illness categories (it was not designed to highlight risk of sepsis) nor did it determine length of stay. Both of these can be determined in future prospective observational cohorts.

## LIMITATIONS

There are limitations to the study design. Admitting or discharging a participant based on the PAGE score is only as appropriate as the original judgement of the clinicians on which the score is based. That is to say we do not know what proportion of admission decisions ultimately prove beneficial to the participant. This is not a limitation specific to our study but a wider consideration for any score that predicts admission. A number of factors influence the decision to admit a participant, and while we have attempted to determine all relevant components there may be sociocultural factors which were missed in the initial derivation work.

## CONCLUSION

For units without the immediate availability of paediatricians the PAGE score can assist staff to determine risk of admission. Cut-off values will need to be adjusted to local circumstances.

**Author affiliations**
[1]CYP@Salford, School of Health and Society, University of Salford, Salford, UK
[2]Emergency Department, North Manchester General Hospital, The Pennine Acute Hospitals NHS Trust, Manchester, UK
[3]Centre for Biostatistics, School of Health Sciences, The University of Manchester, Manchester, UK
[4]Northern Care Alliance NHS Group, Salford Royal Hospital, Salford, UK
[5]Department of Biostatistics, University of Liverpool, Liverpool, UK
[6]SAPPHIRE Group, Health Sciences, University of Leicester, Leicester, UK
[7]Paediatric Emergency Medicine Leicester Academic (PEMLA) Group, Children's Emergency Department, University Hospitals of Leicester NHS Trust, Leicester, UK

**Contributors** Initial study design was by DR, AR, TL, SC, CH and SW with NG, LJB and SB contributing to development. Study data was collected, processed and evaluated by SB, CH and SC. LJB contributed to statistical analysis and review. NG managed the project under the leadership of DR and AR. All authors contributed to both initial and final drafts and agreed to the submission of the final draft.

**Funding** This paper presents independent research funded by the National Institute for Health Research (NIHR) under its Research for Patient Benefit (RfPB) Programme (Grant Reference Number PB-PG-0815-20034).

**Disclaimer** The views expressed are those of the authors and not necessarily those of the NIHR or the Department of Health and Social Care.

**Competing interests** None declared.

**Patient consent for publication** Not required.

**Ethics approval** The West Midlands Research Ethics Committee approved this project (17/WM/0436) on the 20 December 2017.

**Provenance and peer review** Not commissioned; externally peer reviewed.

**Data availability statement** All data requests should be submitted to the corresponding author for consideration. Access to available anonymised data may be granted following review'.

**ORCID iDs**
Laura Jayne Bonnett http://orcid.org/0000-0002-6981-9212

Damian Roland http://orcid.org/0000-0001-9334-5144

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
