## [Reviewer comments · BMJ Open]

ARTICLE DETAILS

TITLE (PROVISIONAL)	An observational cohort study with internal and external validation of a predictive tool for identification of children in need of hospital admission from the emergency department: the Paediatric Admission Guidance in the Emergency department (PAGE) score
AUTHORS	Rowland, Andrew; Cotterill, Sarah; Heal, Calvin; Garratt, Natalie; Long, Tony; Bonnett, Laura; Brown, Stephen; Woby, Steve Roland, Damian

VERSION 1 – REVIEW

REVIEWER	Eyal Klang Mount Sinai Hospital, NY, USA
REVIEW RETURNED	29-Sep-2020

GENERAL COMMENTS	I've read with interest the paper "Development of a multivariable prediction model and scoring tool for identification of children in need of hospital admission from the emergency department: the Paediatric Admission Guidance in the Emergency Department (PAGE) score". This work aimed to develop a risk score for predicting hospital admission in a pediatric population. The study was conducted in three sites. One site served for model training and two sites for validation. The study is comprehensive and well written. Remarks: 1) The developed score includes 14 features. This seems excessive for a manual calculation. Consider either using recursive feature elimination (RFE) to decrease the number of features or alternatively consider deployment as an automatic EHR automatic decision support tool.2) The current study's developed score shows an AUC of ~ 0.78 in the training cohort. The authors suggest using a score's cut-off value of 7. This amounts to a sensitivity of 49% and specificity 88%. This means that 51% of patients usually considered for hospitalization would not be alerted on. Since the authors suggest using the score in settings without pediatricians, consideration should be made regarding this result.3) I suggest adding PPV and NPV values.4) Consider adding "chief complaint" as a feature.5) The current study used logistic regression to calculate the score. Today's "big-data" and machine-learning methods may provide better results. Please look at the work by Yuval Barak-Corren et al. (PMID: 28557729) from 2017. In that work, the authors analyzed ~ 59k pediatric patients for predicting hospital admission. In that study, the authors used a mixed machine-learning model with a logistic
---

	regression-naive Bayes classifier. They showed an AUC 0.91 for the task. Please also look at the work by Roquette et al. (PMID: 32240912) from 2020. The authors analyzed 500k pediatric ED presentations. They used a deep neural network to predict admissions and showed an AUC of 0.892. Current machine-learning literature for adult populations also suggests an AUC ~ 0.9 for predicting hospital admission.
--	---

REVIEWER	D Cheng Royal Children's Hospital Melbourne Australia
REVIEW RETURNED	07-Oct-2020

GENERAL COMMENTS	This manuscript describes validation of a clinical model to predict admission vs discharge in paediatric emergency departments across three institutions. Overall, the manuscript is well written and clear regarding protocol, model validation, results and discussions. Some points to consider:  - it would be good to briefly reference how the variables were selected / chosen - although the full description is given in the referenced protocol paper - Page 22 Lines 12-19 - It is unclear what the last sentence of this paragraph adds. Discussion about HR cutoffs do not link to hospital setting deriving their own cut points; even if both statements by themselves are factually accurate. Suggest rewording the paragraph. The authors do not reference the potential impact of type of problem on the PAGE score. For example, fractures or mental health type conditions may warrant admission, but they may have a low score as the majority of the physiological signs may be normal in these children. How could one
---

VERSION 1 – AUTHOR RESPONSE

Many thanks for the opportunity to revise our manuscript. We have addressed nearly all recommendations or given clear explanations where changes have not been made. We would be keen to engage further if any more changes are required.

VERSION 2 – REVIEW

REVIEWER	Daryl Cheng Royal Children's Hospital Melbourne
REVIEW RETURNED	05-Dec-2020

GENERAL COMMENTS	Authors have addressed reviewer comments in their latest revision. Manuscript reads well with clear conclusions and generalizability for variety of readers and organisations.
--